# Occurrence and Identification of *Aspergillus* Section *Flavi* in the Context of the Emergence of Aflatoxins in French Maize

**DOI:** 10.3390/toxins10120525

**Published:** 2018-12-07

**Authors:** Sylviane Bailly, Anwar El Mahgubi, Amaranta Carvajal-Campos, Sophie Lorber, Olivier Puel, Isabelle P. Oswald, Jean-Denis Bailly, Béatrice Orlando

**Affiliations:** 1Toxalim (Research Center in Food Toxicology), Université de Toulouse, INRA, ENVT, INP-Purpan, UPS, 31027 Toulouse, France; sylvianebailly7@gmail.com (S.B.); anwar.vet2002@gmail.com (A.E.M.); a.carvajalcampos@gmail.com (A.C.-C.); sophie.lorber@inra.fr (S.L.); olivier.puel@inra.fr (O.P.); isabelle.oswald@inra.fr (I.P.O.); 2ARVALIS Institut du Végétal, Station Expérimentale, 91720 Boigneville, France; b.orlando@arvalis.fr

**Keywords:** aflatoxins, *Aspergillus* section *Flavi*, France, maize

## Abstract

Aflatoxins (AFs) are secondary metabolites produced by *Aspergillus* section *Flavi* during their development, particularly in maize. It is widely accepted that AFB1 is a major contaminant in regions where hot climate conditions favor the development of aflatoxigenic species. Global warming could lead to the appearance of AFs in maize produced in Europe. This was the case in 2015, in France, when the exceptionally hot and dry climatic conditions were favorable for AF production. Our survey revealed AF contamination of 6% (*n* = 114) of maize field samples and of 15% (*n* = 81) of maize silo samples analyzed. To understand the origin of the contamination, we characterized the mycoflora in contaminated samples and in samples produced in the same geographic and climatic conditions but with no AFs. A special focus was placed on *Aspergillus* section *Flavi*. A total of 67 strains of *Aspergillus* section *Flavi* were isolated from the samples. As expected, the strains were observed in all AF+ samples and, remarkably, also in almost 40% of AF− samples, demonstrating the presence of these potent toxin producers in fields in France. *A. flavus* was the most frequent species of the section *Flavi* (69% of the strains). But surprisingly, *A. parasiticus* was also a frequent contaminant (28% of the strains), mostly isolated from AF+ samples. This finding is in agreement with the presence of AFG in most of those samples.

## 1. Introduction

Aflatoxins (AFs), and more specifically Aflatoxin B1 (AFB1), are the most dangerous mycotoxins identified to date. Indeed, AFB1 is the most potent natural carcinogenic compound, responsible for hepatocarcinoma in humans and classified by IARC in the group I of carcinogenic molecules for both humans and animals [1,2]. This toxin is also immunosuppressive and has been associated with growth impairment in children [3,4]. In 2010, analysis of several foodborne chemicals by the Chemical and Toxins Disease Task force reported that AFB1 was associated with the highest number of DALYs (death and disability adjusted life years) [5].

AFs are secondary metabolites produced by *Aspergillus* section *Flavi*. *Aspergillus flavus* and *A. parasiticus* are the main species associated with AF contamination of crops. However, in the last decade, section *Flavi* has been studied in depth using molecular tools and several new species have been identified. This section currently comprises 33 different species of which 16 are aflatoxigenic [6,7,8,9,10,11]. These species can be distinguished by subtle morphologic characteristics, gene sequences and by their ability to produce different mycotoxins. For example, *A. flavus*, *A. pseudotamarii* and *A. togoensis*, produce AFs of B type whereas others, including *A. parasiticus*, *A. minisclerotigenes*, *A. korhogoensis*, *A. mottae*, *A. nomius*, and *A. arachidicola*, produce both B and G type AFs. Some species can produce other toxic secondary metabolites such as cyclopiazonic acid (CPA) [12].

Most *Aspergillus* species grow above latitude 25° N & S, with high occurrence between latitudes 26° and 35°, while these species are uncommon in latitudes above 45° [13]. The optimal temperature for growth of *A. flavus* is close to 33 °C [14]. This explains why these fungal species are frequent contaminants of many crops in tropical and subtropical regions where hydrothermal conditions are favorable to both fungal development and the production of toxins [15]. In these geographic areas, AFs are often reported in different kinds of products, mainly maize, spices, peanuts, pistachio nuts and cottonseeds which may be contaminated both before harvest and during storage [16,17].

In Europe, due to its latitude and associated climate, AF contamination was formerly not considered as a real threat to agricultural products produced within the European Union (E.U.) and attention was focused on products imported from third countries [18]. However, global climate change could modify AF distribution and lead to their appearance in areas usually considered as risk free. In 2007, a survey carried out by the European Food Safety Agency (EFSA) demonstrated that AF contamination of agricultural crops within E.U. borders and particularly in southern Europe was an emerging issue [19]. From this date on, several publications reported the presence of AFs in maize in E.U. countries including Romania [20], Italy [21,22], Spain [23], and Croatia [24,25]. In the neighboring country Serbia, contamination of maize harvested in 2012 and used to feed dairy cattle [26] led to the frequent serious contamination by Aflatoxin M1, the hydroxylated metabolite of AFB1, of milk marketed in 2013 [27]. Indeed, 60 to 80% of milk samples analyzed in Serbia were found to be contaminated with levels exceeding E.U. regulatory levels [28,29,30,31]. Moreover, the presence of aflatoxigenic isolates was also reported in other productions such as grapes [32,33].

The contamination of European maize was usually associated with noticeable drought periods [24,34] and it has been demonstrated that both heat and water stress can strongly influence the production of AFs [35]. Recent works that modeled the spread of AFs in Europe showed that global warming could considerably increase the presence of AFs in southern Europe [36]. As an illustration of this trend, the Rapid Alert System for Food and Feed has reported several cases of European maize contamination with AFs since 2013 [37].

In France, no contamination by AFs had been reported before 2015, when exceptional climate conditions in southern France, characterized by high temperatures during maize flowering together with an unusual rainfall deficit, increased the risk of contamination of maize with AFs, as reported in other European countries at the same period [38].

In this context, the aim of this study was to (i) measure the contamination with AFs in maize produced in southern France in 2015 and to (ii) characterize the nature of *Aspergillus* section *Flavi* in these samples. Our results demonstrate that *Aspergillus* section *Flavi* is frequently present in maize exposed to heat and drought and that, in the case of favorable environmental conditions, its development can lead to AF contamination. Surprisingly, *A. parasiticus* appeared to be a frequent contaminant of French maize and its presence leads to the contamination of products with G type AF.

## 2. Results

### 2.1. Climatic Data

Summer 2015 was exceptionally hot and dry in France, with two successive heat waves in July, the maize flowering period, which extends from 1 July to 25 July. Temperatures exceeded 33 °C for 10 to 15 days (Figure 1A), with values 1 to 3 °C above normal in the southwest and eastern half of the country, and differences in sum temperatures ranging from 75 to 145 °C in the same regions (Figure 1A). For the whole summer, the average temperature in France was +1.5 °C above normal, making summer 2015 the second warmest summer after 2003 (anomaly of +3.2 °C) and before 2006 (anomaly of + 1.1 °C).

During the same period, a rainfall deficit of more than 40% on average was also recorded in a large part of the country that lasted from April to the end of July, causing the drying out of the top soil layer. In areas included in the survey, rainfall mostly ranged between 0 and 25 mm during the maize flowering period (Figure 1B). However, local rainfall was sometimes higher. Compared to the average values recorded in France during the 1995 to 2014 reference period, cumulative rainfall in the warmer regions of the study varied between −6 mm and −87 mm (Figure 1B).

### 2.2. Occurrence of Aflatoxins in Maize Samples

#### 2.2.1. Number and Location of Contaminated Samples

For analysis of AF contamination, a total of 118 samples were collected from individual farm fields at harvest (field samples) and 81 more samples were collected from silo sites during the harvest period (silo samples). The silo samples were collected after drying but before storage. They thus corresponded to a mixture of kernels from different fields.

Seven field samples (6%) were found to be contaminated with AFs (AF+ field samples). Twelve silo samples (15%) were also contaminated (AF+ silo samples).

All the contaminated samples were collected in the administrative departments most exposed to heat and drought during the maize flowering period (temperatures > 33 °C during more than seven days and rainfall deficit) (Figure 1 and Figure 2).

Logically, a close link was found between the locations of the contaminated field and silo samples (Figure 2). AF+ silo samples came from seven departments. In three of them, some field samples were also contaminated with AFs. In three other departments, field samples were not contaminated with AFs but *Aspergillus* section *Flavi* were nevertheless detected. In only one department with AF+ silo samples, neither AF+ nor *Flavi* + field samples were found, but this department is located close to contaminated zones.

#### 2.2.2. Types and Levels of AF Contamination

All positive field samples were found to be contaminated by both B and G type AFs. In contrast, AFGs were only found in seven silo samples out of the 12 contaminated ones (58%) (Table 1).

Based on the concentration of toxins, the levels of AFs in three field samples exceeded the E.U. regulation, set at 5 µg/kg for AFB1 and 10 for total AFs in maize to be subjected to sorting or other physical treatment before human consumption or used as an ingredient in foodstuffs. Two of these samples were highly contaminated, with aflatoxin contents of 50.8 and 70 µg/kg, respectively. The last one displayed 15.3 µg/kg AFs. One out of the 12 contaminated silo samples exceeded regulatory levels. However, to comply with E.U. regulations, 2 µg/kg for AFB1 and 4 µg/kg for total aflatoxins in ready-to-eat maize, five samples would require cleaning before being sold.

### 2.3. Fungal Flora of Maize Samples

Samples found to be contaminated with AFs (seven AF+ field samples and 12 AF+ silo samples) were further analyzed to determine fungal contamination and the nature of *Aspergillus* section *Flavi* that were present. To better understand the factors that may have led to contamination by AFs at field level, we also analyzed 24 aflatoxin-free field samples (AF− field samples), collected in the same geographic areas as the contaminated samples, i.e., cultivated under the same climatic conditions. All the fungal flora found in these different types of samples are listed in Table 2 and the complete results of the mycological analyses are given in Appendix A.

#### 2.3.1. AF+ Samples

*Fusarium* was the most frequent genus, in both frequency (100% of samples) and in numbers. *Penicillium* was also systematically present in the samples but the corresponding counts were lower.

*Aspergillus* is a frequently observed fungal genus. The high frequency of *Aspergillus* section *Nigri* is particularly notable. These species, which are ecologically close to *Aspergillus* section *Flavi*, display almost the same pattern as the latter.

*Aspergillus* section *Flavi* were found systematically, in accordance with the presence of AFs in samples. They were present at relatively high levels and the three field samples with the highest AF contamination displayed *Aspergillus* section *Flavi* counts exceeding 2 × 10^4^ CFU/g.

Other sections were also sometimes observed in some silo samples, particularly *Aspergillus* section *Aspergillus*, which are very common contaminants of dried foodstuffs.

*Acremonium* and *Cladosporium*, both typical field contaminants, were also very frequently observed. Other genera were sporadically observed in some samples, for instance, *Verticillium*, *Alternaria*, and *Trichoderma*, common field contaminants. Both *Mucor* and *Rhizopus* were frequently observed in samples but at low counts (Appendix A).

#### 2.3.2. AF− Samples

From a qualitative point of view, the overall pattern of fungal flora observed in AF− samples closely resembled that found in AF+ samples but with lower counts. Surprisingly, *Aspergillus* section *Flavi* were also present in almost 40% of AF− field samples but with moderate counts.

#### 2.3.3. Quantitative Comparison

Quantitative analysis of the mycoflora observed in AF+ vs. AF− field samples pointed to a clear relationship between AF contamination and a global increase in overall fungal contamination, with no change in the nature of fungal genera present (Figure 3).

When considering all AF+ samples (field + silo samples; *n* = 19) vs. AF− samples (*n* = 24), a correlation was found between *Aspergillus* section *Flavi* counts and AF contamination, counts of more than 10^4^ CFU/g being highly correlated with aflatoxin contamination of samples (Figure 4).

### 2.4. Characterization of Aspergillus Section Flavi Strains

A total of 67 strains belonging to the *Flavi* section were isolated. Fifty-five came from AF+ samples (field + silos). Indeed, many samples were contaminated with several morphologically distinct strains as shown in Figure 5. AF+ samples were most often contaminated by three strains per sample but some silo samples were contaminated by 4 (*n* =2) and even 6 (*n* = 2) different strains. In AF− field samples, 12 strains were isolated from the 24 samples analyzed and, in most cases, the samples were contaminated by only one strain.

Identification at the species level using morphological and molecular approaches revealed *Aspergillus flavus* to be the most frequent strain isolated from maize samples, representing 68% of strains (46/67). However, *A. parasiticus* was also frequently identified. Indeed, this species corresponded to 4/12 strains from AF− field samples (33%), 5/21 strains from AF+ field samples (24%), and 10/34 strains from silo samples (30%) as illustrated in Figure 6. As can be seen in Figure 7, the molecular identification clearly confirmed the morphological identification and the location of strains in the *A. parasiticus* clade.

Aside from *A. flavus* and *A. parasiticus*, one strain of *A. minisclerotigenes* and one strain of *A. tamarii* were also identified in silo samples.

The mycotoxin production ability of the strains varied with the species.

Considering strains isolated from AF+ samples, almost 75% of *A. flavus* strains were non-aflatoxigenic and 66% produced CPA. Nine strains (24%) produced both AFB and CPA. Eighty per cent of *A. parasiticus* strains were able to produce B and G AFs, in agreement with the systematic contamination of field maize samples with AFG1.

Considering strains isolated from AF− field samples, 50% of both *A. flavus* and *A. parasiticus* produced B or B and G type AFs, respectively (Figure 6). However, all strains of *A. flavus* were also able to produce CPA and four of them produced both AFB and CPA.

As expected, the *A. minisclerotigenes* strain simultaneously produced AFs B and G and CPA and *A. tamarii* strains produced no AF, but was able to produce CPA.

The toxigenic potential of all the strains is detailed in Appendix A.

Since many strains of *A. flavus* were able to produce CPA, next, this mycotoxin was quantified in the samples. As shown in Appendix A, 28% and 25% of the AF+ field and AF+ silo samples respectively were contaminated by CPA together with AFs, demonstrating the frequent copresence of these toxins. By contrast, none of the AF− field samples were contaminated by CPA.

## 3. Discussion

### 3.1. Impact of Climate on Aflatoxin Emergence in French Maize in 2015

According to Payne [39], the development of *Aspergillus* species and AF production are favored by hot dry weather conditions during the maize growing season. Temperatures suitable for growth of *A. flavus* range from 10 to 12.8 °C to 43 to 48.8 °C with an optimum around 33.8 °C [14]. In France in 2015, the summer was particularly hot, especially during the maize flowering stage and the maximum temperatures recorded were very close to those allowing optimum growth of *Aspergillus* section *Flavi*. In most of the country, these high temperatures were accompanied by a rainfall deficit of more than 40% compared to the average, which dried out the top soil layer and caused heat stress in maize [40], especially during the flowering and silk tanning periods. These exceptional climatic conditions may therefore have enabled the development of *Aspergillus* section *Flavi* and the subsequent AF contamination thus justified a thorough investigation. 

The present survey demonstrated that, in 2015, French maize kernels were contaminated by AFs. These toxins were detected in seven out of 118 field samples (6%) and 12 out of 81 silo samples (15%). All contaminated samples came from departments where the weather conditions at maize flowering were abnormally hot and dry in 2015. In half the cases, AF+ silo samples were found in departments where AF+ field samples were also observed. In other cases, AF+ silo samples came from departments in which no AF+ field samples were found but where the *Aspergillus* section *Flavi* were present. Therefore, contamination of the silos could be the result of contaminated fields that were not analyzed or the result of postharvest development of toxigenic fungi. Indeed, in maize, the postharvest period may play a major role in mycotoxin contamination, especially during pre-storage of grains that still have relatively high water content (~15–20%) before their final drying stage in storage facilities. 

Maximum AFB1 contents of 66 µg/kg were measured in field samples and of 7.2 µg/kg in silo samples. Among the contaminated samples, three field samples and one silo sample exceeded E.U. regulations (5 µg/kg for AFB1 and 10 µg/kg for total AFs in maize to be subjected to sorting or other treatment before consumption) [31]. However, despite dilution during silo filling and the preparation of mixed samples, five silo samples would still require cleaning before sale to reduce AF levels and comply with E.U. regulations (2 µg/kg for AFB1 and 4 µg/kg for total AFs in ready-to-eat maize). Thus, sorting and cleaning of maize kernels appear to be critical points in the maize processing chain. Today, in France, ~80% of maize kernels are cleaned before sale, of which 68% occurs at reception in storage facilities before drying, 20% during storage, and 12% when shipped [41], thereby making it possible to reduce AF content and reach both sanitary and commercial targets. In the future, it would be of great interest to investigate the real impact of this processing step on fungal flora, and if it also reduces *A.* section *Flavi* counts, what is more, this step needs to be accomplished as far upstream as possible, i.e., before the kernels are dried to limit the risk of fungal development and subsequent production of toxins during storage. 

This is the first report of AF contamination of maize kernels produced in France. Contamination has already been reported in other European countries where average climate conditions are more favorable to *Aspergillus* section *Flavi* than conditions in France, including Romania [20] and Italy [42]. More recently, the presence of AFB1 was detected in 57.2% of 180 maize samples collected in 2015 from the main Serbian maize-growing regions, with concentrations ranging from 1.3 to 88.8 μg/kg [38]. Like in France, the authors noted that July 2015 was particularly hot and dry in Serbia with rainfall comparable to that recorded in France and even higher mean temperatures, which may explain why AF contamination was more frequent than in France. These data confirm the emergence of AFs as possible contaminants of European crops and the need to closely monitor the presence of these carcinogenic agents in foods in Europe, which was previously usually considered to be an aflatoxin-free.

Of note, AFG was observed in most contaminated samples, as also recently reported in Serbia [26,38]. The frequent presence of AFG in maize appears to be a new feature of AF contamination of European crops, since, in available studies, only AFB has previously been reported [21,24,25]. Since *A. flavus* is not able to produce AFG, this points to the presence of other aflatoxigenic species in maize, and this was subsequently confirmed by mycological analysis. Our study also demonstrated the possible co-contamination of cereals with AFs and CPA. Such co-contamination has already been reported in mixed feed [43] and in peanuts in Argentina [44] but we found no recent data concerning contamination of cereals. An investigation of the possible interaction between AFs and CPA when ingested simultaneously would be needed to assess the risk of co-contamination of foods [45].

### 3.2. Nature of Aspergillus Section Flavi Responsible for Aflatoxin Contamination of French Maize

Only a few studies describe the mycoflora of European maize but those that do, usually report very similar fungal flora to that we observed, with the frequent presence of *Fusarium*, *Penicillium*, and to a lesser extent, genera such as *Alternaria* and *Cladosporium* [20,21,23]. In the present study, the high prevalence of *Aspergillus* section *Flavi* was also observed but this is not surprising in samples that are being analyzed because of the presence of AFs. Nevertheless, we demonstrated a clear relationship between *A.* section *Flavi* counts and the level of AF contamination, and 10^4^ CFU/g is a possible threshold. We also demonstrated the frequent presence of *Aspergillus* section *Flavi* in AF− field samples. This finding is of major importance since it demonstrates that *A.* section *Flavi* are frequently present in French fields and may therefore be responsible for AF contamination if and when environmental conditions become favorable. This observation is even more important considering that the proportion of toxigenic strains observed in AF− samples was higher than that in AF+ samples. However, counts of *Aspergillus* section *Flavi* were lower in AF− samples than in AF+ samples. Since these samples were collected in areas close to contaminated areas, i.e., grown under the same climatic conditions, it would be now of great interest to compare the agricultural practices being used to identify those that interfere with the growth of the toxigenic *Aspergillus*, as has already been demonstrated for peanuts [46].

The results of our investigation of *Aspergillus* section *Flavi* species were surprising since *A. parasiticus* appeared to be a frequent contaminant of French maize, as confirmed by the presence in many contaminated samples of AFG. Indeed, in most surveys, *A. flavus* is reported to be the main contaminant of maize particularly in Africa [47,48] and USA [49] but also in Europe [50] even if *A. parasiticus* was found in few samples [23,51]. This finding is also in agreement with data from Serbia showing the presence of AFG in maize samples [38]. Since *A parasiticus* requires lower temperatures to colonize substrates, it could be more adapted to the French climate [52].

Among the 67 strains of *Aspergillus* section *Flavi* isolated from maize samples, 31 were aflatoxigenic (46%). This proportion is similar to proportions already reported in strains isolated from maize [23,48,50]. When we examined toxigenic potential as a function of the fungal species, the proportion of toxigenic *A. parasiticus* strains was much higher than the proportion of *A. flavus* with 84 and 30% of aflatoxigenic strains, respectively. This is extremely important due to the relatively high prevalence of *A. parasiticus* isolates that could be responsible for AF contamination of French maize if and when environmental conditions allow their development. The relatively high proportion of non-aflatoxigenic strains of *A. flavus* that are naturally present is also of interest. Indeed, the use of atoxigenic *A. flavus* strains as a biocontrol agent to compete with toxigenic strains has been shown to be a potent strategy [53]. Thus, some of these strains could be tested for their ability to serve as biocontrol agents. Another possible strategy would be to identify agricultural practices able to favor the implantation of these naturally present atoxigenic strains in fields thus giving them a long-term advantage over aflatoxigenic strains that would subsequently limit the risk of recombination [54]. However, many of these non-aflatoxigenic strains produce CPA. Indeed, almost 74% of *A. flavus* strains produced CPA, in some cases leading to the contamination of maize with that mycotoxin. In most cases, we found a high correlation between the counts and toxigenic potential of *A.* section *Flavi* strains isolated from samples and mycotoxin contamination of maize. However, in two cases, we were unable to identify toxigenic strains from AF+ samples despite several platings and isolation of the strains. Alborch et al. [23] already reported similar differences which could be related to the sample drying procedure that led to the inactivation of spores before fungal analysis.

## 4. Conclusions

This work reports, for the first time, the contamination of French maize kernels with AFs. Six percent of analyzed samples taken in fields and 15% of those taken in silos where found to be contaminated, sometimes at high levels, largely exceeding E.U. regulations. This survey also demonstrated that contamination may differ in samples produced under the same climatic conditions but using different agricultural practices. Finally, it also highlighted the fact that AFB1 contamination was not only associated with *A. flavus* but with the presence of other species, particularly *A. parasiticus*, leading to contamination with both B and G type AFs. All these data confirm the emergence of aflatoxins in France and the need for close monitoring of these contaminants in susceptible crops throughout the territory. 

## 5. Materials and Methods

### 5.1. Meteorological Data

The weather variables used to qualify the atypical 2015 growing season are based on two daily parameters: temperature and rainfall. These variables were calculated from spatialized climatic data from nearly 700 weather stations distributed throughout metropolitan France [55].

### 5.2. Sampling

#### 5.2.1. Sampling Strategy

All the samples were collected during the 2015 maize growing season, at harvest.

In the first round, 118 field samples and 81 samples collected from silos were analyzed for AF contamination. The field samples were collected in 35 French administrative departments and the silo samples in 33 departments (Figure 2). Following this first round of analysis, samples found to be contaminated with aflatoxins (7 field samples and 12 silo samples) were further analyzed to identify fungal contamination and the nature of *Aspergillus* section *Flavi* present. To better understand the factors that may have led to aflatoxin contamination of the samples, the second round of analyses was undertaken using also 24 uncontaminated field samples collected in the same departments as contaminated ones, i.e., in areas where climatic conditions represented a high risk of aflatoxin synthesis. All the samples in the second-round of analyses were also analyzed for cyclopiazonic acid contamination.

#### 5.2.2. Sample Collection

##### Field Samples

A total of 118 farm fields planted with maize were sampled: at harvest, the farmer was asked to prepare samples respecting the following recommendations. (a) Avoid sampling the margins of the field, (b) avoid static sampling of grain, and (c) sample moving grains during three different periods of emptying of the combine harvester. In this way, three different subsamples, each weighing at least 1 kg, were manually collected from the moving grains during harvest. These three subsamples were then combined to obtain a 3 kg final sample from each farm field.

##### Silo Samples

A total of 225 samples were collected in silos belonging to storage companies, cooperatives and private merchants located in 33 French administrative departments. At harvest time, three elementary dried samples were taken before storage at different sampling dates (beginning, middle, and end of harvest period). The resulting 225 samples were therefore representative of the different silos. The elementary samples were then mixed to prepare 81 mixed samples representative of each department, each weighing at least 3 kg. AF contents were measured in these 81 mixed samples.

#### 5.2.3. Sample Preparation for Analysis

All the grain samples were cleaned with a laboratory cleaner and separator (MINI-PETKUS 100 and 200, PETKUS Technologie GmbH, Rohr, France) to remove all impurities from the kernels. Then, 1.5 kg of cleaned and homogeneous sample was taken for analysis and ground in a laboratory hammer mill fitted with a 1 mm screen (TITAN 2000, F.A.O., Vitré, France).

### 5.3. Aflatoxin and Cyclopiazonic Acid Quantification

AFs B1, B2, G1, and G2 were analyzed by liquid chromatography-tandem mass spectrometry in a French accredited laboratory. The limits of detection for AFB1, B2, G1, and G2 were 0.1, 0.1, 0.12, and 0.25 µg/kg, respectively, the corresponding limits of quantification were 0.25, 0.25, 0.25, and 0.5 µg/kg. For CPA, the limit of detection was 5.0 µg/kg, and the corresponding limit of quantification was 10.0 µg/kg.

### 5.4. Fungal Count and Identification

Fungal counts were made according to ISO 7954 norm [56]. In brief, 20 g of ground sample was mixed with 180 ml of 0.05% Tween 80 for 2 min in a Waring blender and then placed on a horizontal shaking table at 220 rpm for one hour. Decimal dilutions were prepared in 0.05% Tween 80 and 100 µl of each dilution were plated on both MEA medium and salted MEA (MEA + 6% NaCl). The latter medium was used to identify xerophilic species and to limit the development of mucorales which can prevent correct counting and identification of species with low growth rates. Fungal colonies were counted after three days of culture at 25 °C and confirmed after five days. The limit of detection for the fungal count was 10 CFU/g of sample. The colonies were then identified according to Pitt and Hocking [14] and Samson et al. [57]. *Aspergillus* section *Flavi* strains were isolated from plates by several platings on MEA and salted MEA.

### 5.5. Characterization of Aspergillus Section Flavi

#### 5.5.1. Morphological Identification

*Aspergillus* section *Flavi* strains were identified at the species level through macroscopic and microscopic examination after five and seven days of culture at 25 °C on MEA and salted MEA according to Pitt et al. [58] and Varga et al. [7].

#### 5.5.2. Molecular Identification

The molecular identification was performed on all *Aspergillus* section *Flavi* strains which did not display typical *A. flavus* morphological features and on two *A. flavus* strains as control. Identification was performed by internal transcribed spacer (ITS) beta-tubulin (*benA*) and calmodulin (*cmdA*) gene amplification and sequencing. The strains were cultured in yeast extract sucrose (YES) broth and placed on a shaking incubator at 160 rpm at 27 °C for three days. Genomic DNA was isolated from mycelia as previously described [59]. Primers used for molecular identification are listed in Table 3. PCR reactions were carried out in a GeneAmp PCR 2700 thermocycler (Applied Biosystems, Foster City, CA, USA). PCR products were purified with a GenElute PCR Clean-Up Kit (Sigma-Aldrich, Saint-Quentin Fallavier, France) and sequenced on an ABI3130XL sequencer (Applied Biosystems, Foster City, CA, USA) using the dye terminator technology. PCR products were sequenced in both directions.

New sequences were blasted against NCBI database, and deposited in GenBank. Accession numbers of *A. parasiticus* strains are listed in Table 4.

The species identification was confirmed by phylogenetic analyses. Data were assembled, aligned and trimmed using ClustalW in BioEdit v7.0.5 [60], and checked. In order to reduce indels and to optimize nucleotide identities, regions with multiple gaps were aligned. The *ITS* gene was analyzed independently, whereas, *benA* and *cmdA* genes were concatenated using Mesquite v3.2. (Mesquite Software, Austin, TX, USA) [61]. The best-fit nucleotide substitution model for *ITS* was calculated using jModelTest, and resulted on TIM2 + I + G substitution model. For concatenated data, the best-fit nucleotide substitution model and partition scheme were calculated using PartitionFinder v2.0.0 (Australian National University, Canberra, Australia) [62]; one partition was suggested under K80+G substitution model.

Bayesian analyses were performed using MrBayes v3.2 [63]; four independent runs, each one with four chains, were carried out for 10^7^ generations. Sampling was performed every 10^3^ generations. For each analysis, we checked that the average standard deviation of split frequencies among chains were near ≤0.01, and the potential scale reduction factor (PSFR) to 1. From the total number of trees per run, 25% were used as burn-in; and the remaining trees were used to calculate the posterior probabilities (PP), based on the 50% majority rule consensus tree by Tracer v1.6 [64]. Visualization and edition of phylogenetic trees were performed on FigTree v1.4.2 (Institute of Evolutionary Biology, University of Edinburgh, Edinburgh, Scotland) [65].

#### 5.5.3. Mycotoxigenic Potential of Isolates

To assess the toxigenic potential of *Aspergillus* section *Flavi* strains, a spore suspension was prepared from a 7-day culture and 100 spores were centrally inoculated on MEA and cultured for seven days at 25 °C. After this incubation period, highly sporulating cultures were analyzed for toxin production as already described [66,67].

## Figures and Tables

**Figure 1 toxins-10-00525-f001:**
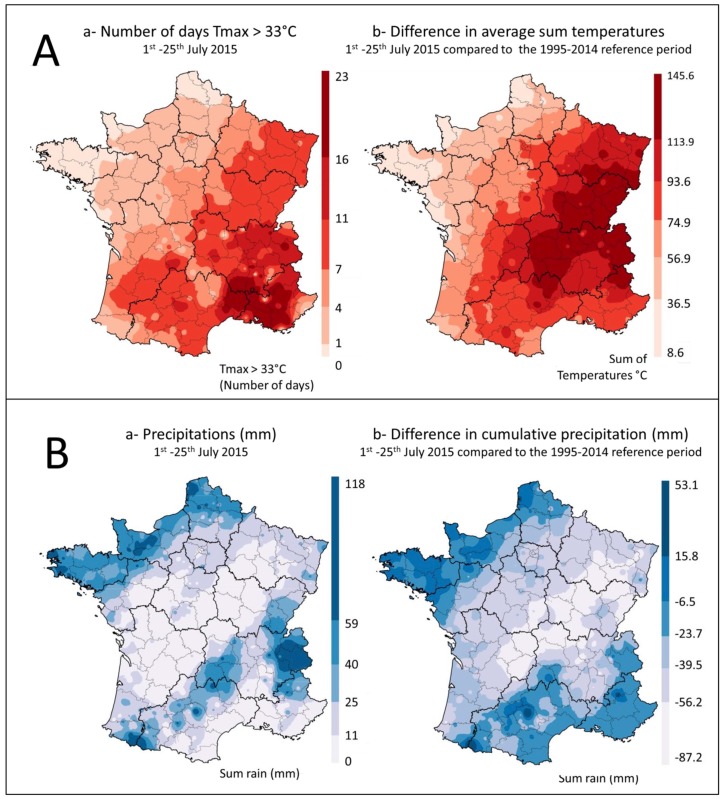
Temperatures and rainfall during the maize flowering period in France. (**A**): Temperatures (**a**) number of days with T_max_ > 33 °C between 1 July and 25 July and (**b**) comparison of the sum of daily temperatures between 1 July and 25 July, with the average for the 1995–2014 period. (**B**): Precipitation (**a**) Sum of precipitation between 1 July and 25 July (mm) and (**b**) difference in cumulative rainfall between 1 July and 25 July 2015 and the average for the same period between 1995 and 2014 years (mm).

**Figure 2 toxins-10-00525-f002:**
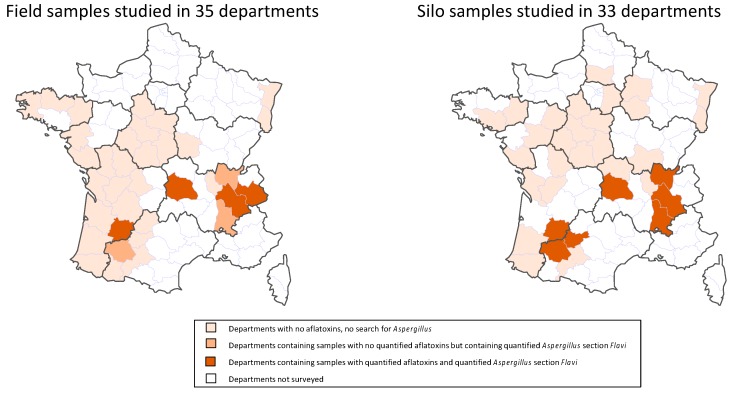
Geographic distribution of field and silo samples according to their contamination status with AFs and *Aspergillus* section *Flavi*.

**Figure 3 toxins-10-00525-f003:**
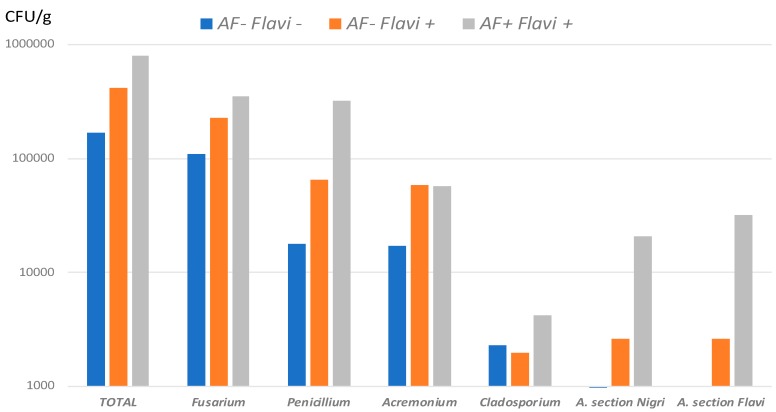
Fungal flora found in field samples. Mean counts of total mycoflora and of more frequent genera in AF− *Flavi*− (*n* = 15), AF− *Flavi*+ (*n* = 9) and AF+ *Flavi*+ (*n* = 7). Results are expressed in CFU/g.

**Figure 4 toxins-10-00525-f004:**
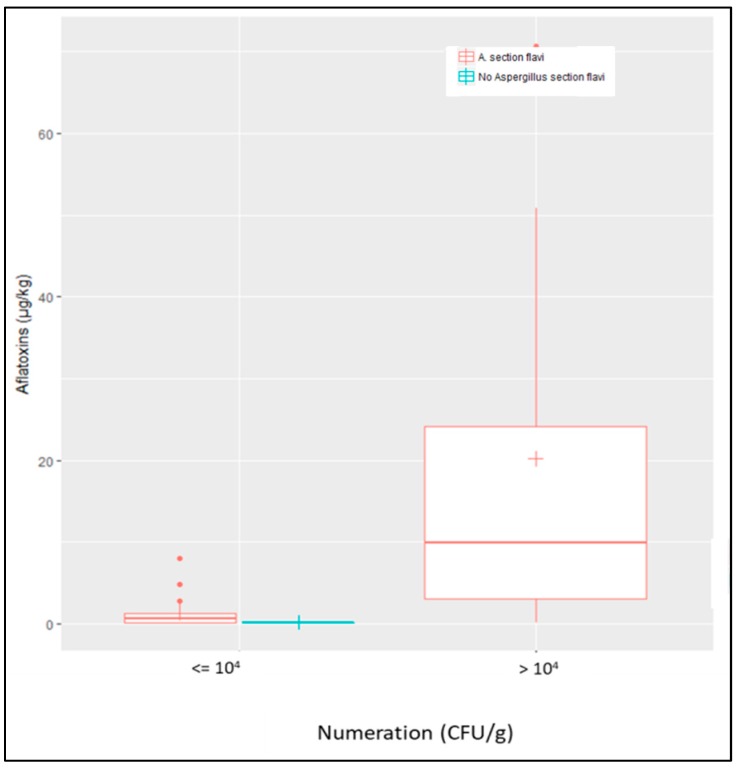
Correlation between counts of *Aspergillus* section *Flavi* and contamination by AFs. *Aspergillus* section *Flavi* were systematically found in AF+ samples as shown by red boxplots and no sample was contaminated by AFs in the absence of *A.* section *Flavi* (blue boxplot). Counts of *A.* section *Flavi* higher than 10^4^ CFU/g were highly correlated with AF contamination of samples, whatever their origin (field or silo).

**Figure 5 toxins-10-00525-f005:**
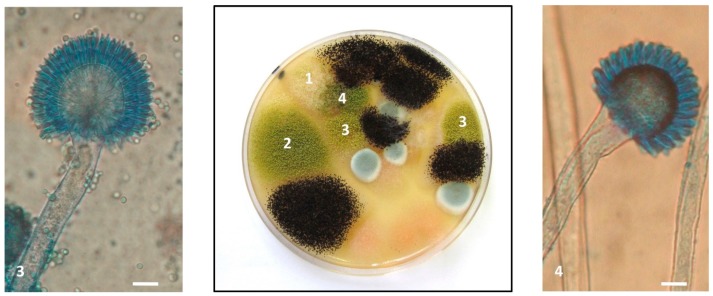
Presence of four different *Aspergillus* section *Flavi* strains in a single sample. The Petri dish in the figure corresponds to the 10^−3^ dilution of sample M638 and was observed after 7 days of incubation at 25 °C on malt extract agar (MEA). Five colonies corresponding to four morphologically distinct strains of *Aspergillus* section *Flavi* were present and were isolated for identification and characterization. 1: *A. flavus* M638a; 2: *A. flavus* M638e; 3: *A. flavus* M638g; and 4: *A parasiticus* M638b. The three strains of *A. flavus* displayed very different toxigenic potential. In the left panel, microscope image (×400) of *A. flavus* M638g characterized by a biseriate conidial head, a globose vesicle, radiate on ¾ and a rough conidiophore. In the right panel, microscope image (×400) of *A. parasiticus* M638b with uniseriate head, a sub-globose vesicle and a smooth conidiophore. Scale bar: 10 µm.

**Figure 6 toxins-10-00525-f006:**
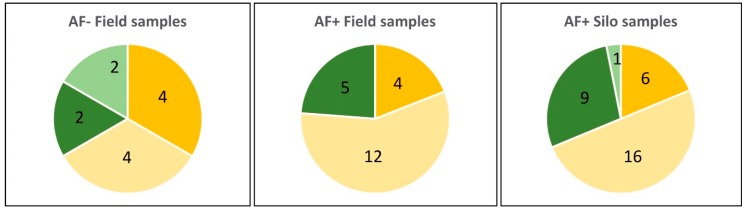
Number of *A. flavus* and *A. parasiticus* strains in samples as a function of their origin and contamination status with AFs. Light green: nontoxigenic *A. parasiticus* strains; dark green: toxigenic *A. parasiticus* strains; light yellow: nontoxigenic *A. flavus* strains; dark yellow: toxigenic *A. flavus* strains.

**Figure 7 toxins-10-00525-f007:**
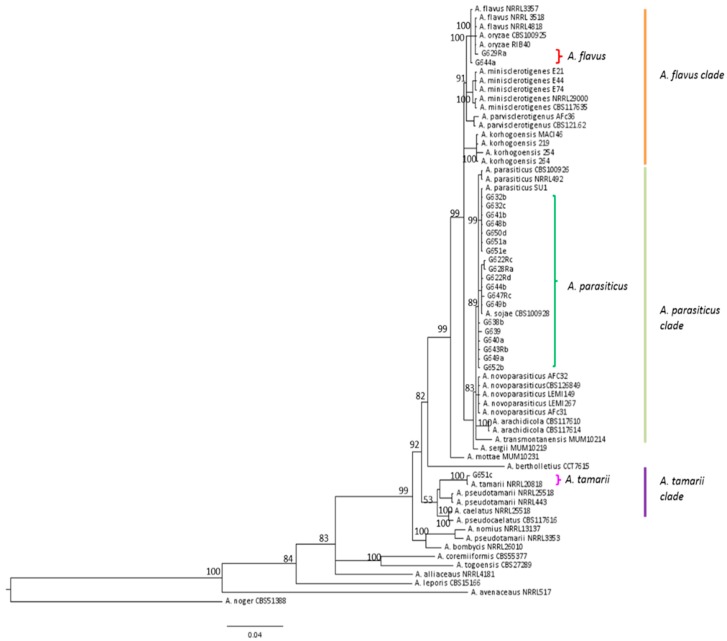
Phylogenetic trees showing the location of all *A. parasiticus*, two typical *A. flavus* and the only *A. tamarii* strains isolated from French maize samples. *benA* and *cmdA* genes were concatenated using Mesquite v3.2. and phylogenic tress were created on FigTree v1.4.2.

**Table 1 toxins-10-00525-t001:** AF content of contaminated maize samples.

Origin	Contaminated Samples (%)	Aflatoxin Content (µg/kg)				
		AFB1	AFB2	AFG1	AFG2	Total AFs
		1.1	ND	0.4	ND	**1.5**
		0.1	ND	0.2	ND	**0.3**
		4.0	0.4	9.7	1.2	**15.3**
**FIELD**	**7 samples/118**	0.3	ND	2.0	0.3	**2.6**
	6%	0.7	ND	0.5	ND	**1.2**
		20.4	2.1	24.8	3.5	**50.8**
		66.0	3.1	0.9	ND	**70**
		0.1	ND	0.7	ND	**0.8**
		3.2	0.2	4.5	ND	**7.9**
		0.6	ND	ND	ND	**0.6**
		3.3	0.2	ND	ND	**3.5**
		2.4	0.2	ND	ND	**2.6**
**SILOS**	**12 samples/81**	1.4	ND	0.1	ND	**1.5**
	15%	3.8	0.2	0.8	ND	**4.8**
		0.6	ND	ND	ND	**0.6**
		1.9	0.2	3.9	0.6	**6.6**
		7.2	0.3	5.6	0.2	**13.3**
		0.4	ND	0.1	ND	**0.5**
		0.7	ND	ND	ND	**0.7**

ND: not detected.

**Table 2 toxins-10-00525-t002:** Total fungal flora found in maize samples.

Mycoflora	Mean Fungal Load (CFU/g) (Mini–Maxi) *% Contaminated Samples*		
	AF− Field Samples (*n* = 24)	AF+ Field Samples (*n* = 7)	AF+ Silo Samples (*n* = 12)
Total flora	2.3 × 10^5^	7.9 × 10^5^	2 × 10^5^
	(1.2 × 10^4^–1.2 × 10^6^)	(6.9 × 10^4^–2.5 × 10^6^)	(1.4 × 10^4^–7.7 × 10^5^)
*Fusarium* sp.	1.6 × 10^5^	3.5 × 10^5^	1.7 × 10^5^
	(10^3^–5 × 10^5^)	(10^4^–2 × 10^6^)	(10^4^–7 × 10^5^)
	*100%*	*100%*	*100%*
*Penicillium* sp.	3.5 × 10^4^	3.2 × 10^5^	6.1 × 10^3^
	(2 × 10^2^–4 × 10^5^)	(5 × 10^3^–2 × 10^6^)	(5 × 10^2^–3 × 10^4^)
	*100%*	*100%*	*100%*
*Acremonium* sp.	3.3 × 10^4^	5.7 × 10^4^	1.1 × 10^4^
	(0–3 × 10^5^)	(0–2 × 10^5^)	(0–3 × 10^4^)
	*67%*	*86%*	*58%*
*Cladosporium* sp.	2.2 × 10^3^	4.2 × 10^3^	1.3 × 10^2^
	(0–10^4^)	(0–2 × 10^4^)	(0–10^3^)
	*79%*	*86%*	*25%*
*Aspergillus* sp.	2.2 × 10^3^	5.3 × 10^4^	9.2 × 10^3^
	(0–2 × 10^4^)	(2 × 10^3^–1.7 × 10^5^)	(10^2^–4 × 10^4^)
	*54%*	*100%*	*100%*
*Aspergillus* section *Nigri*	10^3^	2.1 × 10^4^	5.8 × 10^2^
	(0–2 × 10^4^)	(10^2^–7 × 10^4^)	(0–4 × 10^3^)
	*46%*	*100%*	*83%*
*Aspergillus* section *Flavi*	9.9 × 10^2^	3.2 × 10^4^	6.2 × 10^3^
	(0–2 × 10^4^)	(6 × 10^2^–10^5^)	(10^2^–3 × 10^4^)
	*38%*	*100%*	*100%*

**Table 3 toxins-10-00525-t003:** Sequence of the primers used for molecular identification of *Aspergillus* section *Flavi* isolates.

Gene	Gene Name	Length (bp)	Primers		Sequence (Nucleotides: 5’→3’)
			Forward	Reverse	
*ITS* (4–5)	Internal transcribed spacer	300–330	ITS5		5′-GGAAGTAAAAGTCGTAACAAGG
				ITS 4	5′-TCCTCCGCTTATTGATATGC
*benA-2*	ß-tubulin	1125	benA 2a		5′-GGTAACCAAATCGGTGCTGC
				benA 2b	5′-ACCCTCAGTGTAGTGACCCTTGGC
*cmdA*	Calmodulin	543	Cmd5		5′-CCGAGTACAAGGAGGCCTTC-3’
				Cmd6	5′-CCGATAGAGGTCATAACGTGG-3’

**Table 4 toxins-10-00525-t004:** Accession numbers deposited in GenBank of *A. parasiticus* isolated from French maize samples.

Species and Strain Number	Accession Number		
*A. parasiticus*	*ITS*	*cmdA*	*benA*
G622Rc	MK165710	MK165730	MK172077
G622Rd	MK165711	MK165731	MK172078
G628Ra	MK165712	MK165732	MK172079
G632b	MK165713	MK165733	MK172080
G632c	MK165714	MK165734	MK172081
G638b	MK165709	MK165735	MK172082
G639	MK165726	MK165736	MK172083
G640a	MK165715	MK165737	MK172084
G641b	MK165716	MK165738	MK172085
G643Rb	MK165717	MK165739	MK172086
G644b	MK165718	MK165740	MK172087
G647Rc	MK165719	MK165741	MK172088
G648b	MK165720	MK172072	MK172089
G649a	MK165708	MK165742	MK172090
G649b	MK165721	MK165743	MK172091
G650d	MK165722	MK165744	MK172092
G651a	MK165723	MK165745	MK172093
G651e	MK165724	MK172073	MK172094
G652b	MK165725	MK172096	MK172095

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
