# Peer review of "Occurrence and Identification of Aspergillus Section Flavi in the Context of the Emergence of Aflatoxins in French Maize"

_toxins, 2018, doi:10.3390/toxins10120525_

Round 1
Reviewer 1 Report
The paper "Occurrence and identification of Aspergillus section Flavi
in the context of the emergence of aflatoxins in French maize" shows an
interesting report on mycotoxin (aflatoxins) contamination and
Aspergillus spp. infection occurring in maize fields in France in 2015.
The paper is well written, easy to be followed by the reader and English
language does not need any revision. From a scientific point of view
both data collection and analysis are well done, fungal strains
adeguately identified both by a morphological and a molecular approach
and there are enough information about the mycotoxigenic potential of
these strains.
In my opinion the ms does not need modifications before its publication just apart a very minor suggestion I would like to give to authors concerning a result that could be of interest if discussed deeply in the text. In results authors reported that almost 75% of A. flavus strains are non-toxigenic (page 9, line 211-212). This is a very interesting point expecially in the view of the control (or prevention) of aflatoxigenic strains contamination, this last very strightly connected with climatic condition, as deeply discussed by authors. Non-aflatoxigenic strains are well known as potential tool to compete with toxigenic ones and can be used to prevent mycotoxin contamination (see Ehrlich, 2014 and Sarrocco and Vannacci, 2018). I would like to invite authors to discuss about this point also in view of a prevention strategy when they talk about the aflatoxigenic potential of A. fluvus strains (page 11, line 306-310).
Author Response
The authors thank the reviewer for its very interesting comments. Indeed, the presence of non-aflatoxigenic strains is important as a possible strategy to control aflatoxigenic ones and subsequent contamination of crops. Two ways could be used:
- Testing some non-aflatoxin producing strains as potent biocontrol agent. Since they are already present in French fields, they could easily adapt to climate conditions. However, as noted by Ehrlich (2014), the question of possible recombination with toxigenic strains still remains.
- Identify agricultural practices that may favor the development/implantation of such non-toxigenic strains in French fields by giving them a long-term competitive advantage on toxigenic ones and subsequently limit the risk of recombination.
These two possibilities were discussed in the revised version of the manuscript (page 11, lines 705-712 of the track changes version) and references by Erhlich (2014) and Sarrocco and Vanaacci (2018) were added (references number 53 and 54 respectively).
Reviewer 2 Report
The manuscript entitled "Occurrence and identification of Aspergillus section Flavi in the context of the emergence of aflatoxins in French maize" is a survey on the presence of Aspergillus Flavii in mais in France. The topic is within the scope of the Journal and it refers to a specific context which is of great interest. There are some minor remarks, e,g. please define fully the acronyms at the first time when they are used in the text (see: E.F.S.A. instead of EFSA, etc.). It should be useful to make a short comparison or comment about other crops (not only milk) contaminated by the same metabolite, as an example the following paper by: Mikusova P, Sulyok P. Santini A, Srobarova A. Aspergillus spp. and their secondary metabolite production in grape berries from Slovakia, Phytopathologia Mediterranea, (2014), 53, 2, 109-114 should be cited and added to the reference list.
Also the manuscript by: Ketney O., Santini A., Oancea. (2017), Recent aflatoxin survey data in milk and milk products: A review. International Journal of Dairy Technology, 70 (3), 1-12; and the one by Mikusova, P.; Ritieni, A.; Santini, A.; Juhasova, G.; Srobarova, A., Contamination by moulds of grape berries in Slovakia, Food Additives & Contaminants, Part A: Chemistry, Analysis, Control, Exposure & Risk Assessment (2010), 27(5), 738-747, must be cited and will add information to the Introduction section of the manuscript widening the area of interest. Please cite them in the text and in the Reference list. The experimental part is properly assessed and described. The conclusions are clear and properly assessed. English is fine, moderate English spelling/phrasing would be advisable.
Author Response
The authors thank the reviewers for its comments on the article.
As recommended, the acronym EFSA was written fully in the text (page 2, line 73).
The review on AFM1 presence in milk from different origin, including Serbia in 2013 was added in the text (page 2, line 79). Some other references were added in the introduction section to expand the area of interest and particularly the papers by Mikusava et al. (2010) and Mikusava et al. (2014) demonstrating the presence of toxigenic strains of A. flavus in grape berries were added (Page 2, lines 80-81).
The text was also edited by a native English professional translator to improve phrasing.
Reviewer 3 Report
Dear Authors
You pretend that this study reports for the first time the contamination of French maize with AFs.
If so, congratulations
Please check eventually previous reports
Author Response
The authors thank the reviewer for his congratulations. Indeed, we think that it is the first report demonstrating the presence of aflatoxins in French maize kernels. Some previous publications reported the presence of AFM1 at low levels in French milk samples (or cheeses produced in France) (Boudra et al., 2007; Piva et al., 1988). But in these publications, there was no direct relationship with the origin of cereals that may have been used to feed the animals. Nevertheless, it is interesting to note that the study by Boudra et al was done on milk produced in 2003, and exceptionally hot and dry year in France which could explain these results. But maize was not analyzed there.
Nevertheless, to indicate more precisely the originality of our findings, we indicate in the “key contribution section” as well as in the ”discussion” and “conclusion” that it is the first description of the presence of AFB1 in “French maize kernels” (page 1, line 22; page 10, line 486 and page 12, line 710 of the track changes version)
The text was also edited by a Native English professional translator to improve phrasing.